# A Prospective Study on Risk Prediction of Preeclampsia Using Bi-Platform Calibration and Machine Learning

**DOI:** 10.3390/ijms251910684

**Published:** 2024-10-04

**Authors:** Zhiguo Zhao, Jiaxin Dai, Hongyan Chen, Lu Lu, Gang Li, Hua Yan, Junying Zhang

**Affiliations:** 1Hangzhou Research Institute, Xidian University, Hangzhou 311231, China; zgzhao@stu.xidian.edu.cn; 2School of Computer Science and Technology, Xidian University, Xi’an 710071, China; 3School of Telecommunications Engineering, Xidian University, Xi’an 710071, China; 20012100082@stu.xidian.edu.cn; 4School of Medicine, Northwest University, Xi’an 710127, China; 202233091@nwu.edu.cn; 5National Engineering Research Center for Miniaturized Detection, Xi’an 710127, China; lulu@lifegen.com.cn (L.L.); ligang@lifegen.com.cn (G.L.)

**Keywords:** preeclampsia risk prediction, random forest algorithm, multilayer perceptron, bi-platform calibration, data imbalance problem

## Abstract

Preeclampsia is a pregnancy syndrome characterized by complex symptoms which cause maternal and fetal problems and deaths. The aim of this study is to achieve preeclampsia risk prediction and early risk prediction in Xinjiang, China, based on the placental growth factor measured using the SiMoA or Elecsys platform. A novel reliable calibration modeling method and missing data imputing method are proposed, in which different strategies are used to adapt to small samples, training data, test data, independent features, and dependent feature pairs. Multiple machine learning algorithms were applied to train models using various datasets, such as single-platform versus bi-platform data, early pregnancy versus early plus non-early pregnancy data, and real versus real plus augmented data. It was found that a combination of two types of mono-platform data could improve risk prediction performance, and non-early pregnancy data could enhance early risk prediction performance when limited early pregnancy data were available. Additionally, the inclusion of augmented data resulted in achieving a high but unstable performance. The models in this study significantly reduced the incidence of preeclampsia in the region from 7.2% to 2.0%, and the mortality rate was reduced to 0%.

## 1. Introduction

Preeclampsia (PE) is a complex and relatively common idiopathic multisystem disorder that poses significant risks during pregnancy. It is a major cause of maternal morbidity and infant mortality, accounting for approximately 20% of global maternal deaths and 15% of global preterm births [1,2]. Each year, there are about 8.5 million cases of PE worldwide, leading to over 70,000 maternal deaths [3]. It is the second leading cause of death among pregnant women globally.

Early prediction of PE risk before 16 weeks of gestation is crucial in preventing eclampsia, a serious complication of pregnancy characterized by convulsions. This prediction can improve obstetric care, reduce healthcare costs, and lead to significant savings in the health sector.

As one of the largest regions in China, Xinjiang is characterized by vast desert areas and mountainous terrain and is home to ethnic groups such as Uyghurs, Han, Kazakhs, and many others. This geographical diversity creates significant challenges in regard to healthcare accessibility, particularly for rural communities. The worldwide incidence of PE is about 2–8%, although it may be as high as 9.1% in the region, the highest incidence in China due to the unique eating habits and lifestyle of the different ethnic groups within the country.

This study aims to construct PE risk prediction models specifically tailored for the population in Xinjiang, China.

### 1.1. Aim of This Study

The cause of PE is unknown [1], and diagnosis of PE is based on a combination of clinical signs and symptoms, including high blood pressure (hypertension), proteinuria (the presence of protein in the urine) [4], etc., and placental growth factor (PlGF), which has been confirmed to be the best biomarker and the metric of PE [5,6,7] that is recommended by The International Federation of Obstetrics and Gynecology, FMF series studies, and FIGO Guidelines (2019) most suitable for diagnosing this condition [8,9] (FMF, the Fetal Medicine Foundation; FIGO, the International Federation of Gynecology and Obstetrics).

PlGF can be detected via Single-Molecule Array (SiMoA), which is performed on a highly sensitive protein marker detection platform based on the Poisson distribution principle and single-molecule technology [10]. It can also be detected in peripheral blood using the ROCHE Elecsys Test [6]. The two platforms are widely used; however, they differ in terms of measurement range, sensitivity, and specificity [2].

The aim of this study is to construct models for predicting PE risk and early PE risk using clinical symptoms and the PlGF level, which is measured using either the SiMoA or the Elecsys platform, for the population of Xinjiang, China.

### 1.2. Literature Review

Artificial intelligence (AI) techniques have become a tool for efficient, convenient, and intelligent prediction of the risk of PE [2].

In early screening and diagnosis of PE, Sirinat Wanriko et al. proposed a method for developing a prediction model for risk assessment of pregnancy-induced hypertension using a machine learning approach [11], where data were amplified using the Synthetic Minority Oversampling Technique (SMOTE) algorithm [12] and seven machine learning algorithms were experimentally compared with the result, which revealed that Random Forest (RF) provides the best prediction performance. Ivana Marić et al. constructed a PE risk prediction model with an elastic network trained by a gradient boosting algorithm [13]. Shilong Li et al. used the gradient boosting tree algorithm to model data on features in women’s electronic medical records (EMRs) during three major pregnancy time periods (antenatal, intrapartum, and postpartum) [14].

In a study using PlGF, Herdiantri Sufriyana et al. proposed an M5P (M5 model trees) algorithm consisting of a decision tree and four linear models with different thresholds for predicting PE and intrauterine growth restriction [15] with features including PlGF. Xu Qi et al. proposed a method of predicting hypertensive disorders in pregnancy (HDP) based on PlGF [15], which was aimed at studying the effect of the HDP model with or without PlGF, with the result that adding PlGF improved model accuracy. And Bernat Serra et al. proposed an effective multivariate Gaussian distribution model using maternal factors [16], early PlGF determination, and biophysical variables for early-onset PE screening in routine care settings.

Recent studies have focused on using broader biomarkers, utilizing deep learning methods and integrating genetic factors for PE risk prediction. For example, Garrido-Giménez and his colleagues [17] combined sFlt-1, PlGF, NT-proBNP, and uric acid as biomarkers for a PE predictive model using a machine learning method; Wang and his colleges proposed a single-cell transcriptome-based PE risk assessment using an ensemble machine learning framework [18]. Bennett et al. [19] utilized extensive data sources, including the Public Use Data Files (PUDFs) for Texas, the Magee Obstetric Medical and Infant (MOMI) database, and the Oklahoma PUDF in order to predict early PE risk using machine learning and the cost-sensitive deep neural network (CSDNN) method [20]. By incorporating clinical and genetic/omics data into predictive models, personalized PE risk assessment can be improved, leading to higher predictive accuracy [21]. Another direction is to reveal potential PE-related characteristic genes for understanding the mechanism and pathway of PE [22,23,24].

A review on using machine learning and deep learning models for PE risk prediction is given in [25].

To the best of our knowledge, no system has been developed that allows for the prediction of the risk of PE based on PlGF levels measured from either the SiMoA or Elecsys platform. Additionally, the importance of imputing missing data in the context of PE risk prediction has not been adequately addressed in the relevant literature.

### 1.3. Contributions of This Work

In this paper, we propose a machine learning-based approach and construct models for PE risk prediction and early prediction, allowing either the SiMoA or Elecsys platform to be used for measuring PlGF levels.

By selecting quality samples of PE cases and normal controls from more than tens of thousands of samples in the area, the approach showcases innovations and contributions that are significant:(1)A novel missing data imputation method is proposed where multiple strategies are adopted for adapting training data and test data, and independent features and dependent feature pairs. Additionally, a novel PlGF calibration model is established from the excessively small calibration data in this study by training multilayer perceptrons (MLPs) and selecting the one with the median performance for reliably calibrating PlGFs measured by SiMoA and the Elecsys platform.(2)Typical machine learning algorithms such as MLP, Support Vector Machine (SVM), RF, XGBoost, and AdaBoost (with output thresholding for addressing the data imbalance problem) are compared for the best-performing model for PE risk prediction. The result shows that the RF model is the best model.(3)RF models trained on various datasets, such as mono-platform vs. bi-platform data, early pregnancy vs. early plus non-early pregnancy data, and real vs. real plus augmented data, are compared. The results show that the two mono-platform datasets combined can improve PE risk prediction performance, while the non-early pregnancy data can enhance the limited early pregnancy data for better early prediction performance. Additionally, using SMOTE-based data augmentation for model training can lead to virtually high but not stable performance.

The prospective observations based on the models of this study showed that the incidence of PE in hospitalized pregnant women in the district of Xinjiang was reduced from 7.2% to 2.0%, and the mortality rate was reduced to 0.

## 2. Results

As a binary classification problem, PE risk prediction performance is evaluated by error rate, true positive rate (TPR, also called sensitivity and recall), false positive rate (FPR, which is one minus specificity), F1 score, area under receiver operating curve (AUC_ROC), and area under the precision–recall curve (AUC_PRC), which are all based on the confusion matrix. Some may notice the true negative rate (TNR) and false negative rate (FNR), which can be simply derived from TPR and FPR: TNR = 1-FPR, and FNR = 1-TPR. Among all the performance indices, AUC_ROC and AUC_PRC are considered to be robust in the face of imbalanced distributions of positive and negative samples.

To understand the generalization performance of a trained model, we conduct 100 rounds of 10-fold cross-validation to assess not only the median performance but also the performance deviation, which represents the stability of the median performance. We use multiple rounds of 10-fold cross-validation to obtain more reliable performance. Median performance, rather than mean performance, is chosen for its robustness against performance outliers, while the deviation is calculated from the median. In addition, for F1 score, which is defined as the harmonic mean of precision and recall, macro-F1 is based on arithmetic mean of precisions and recalls, and micro-F is based on the arithmetic mean of confusion matrices.

### 2.1. Selecting Prediction Model

Five typical machine learning algorithms for comparison are MLP [26], SVM [27], XGBoost [28], AdaBoost [29], and RF [30,31]. These algorithms encompass a spectrum of machine learning paradigms. MLP represents a neural network-based approach, SVM is grounded in geometric principles, RF constitutes an ensemble method reliant on decision trees, and both XGBoost and AdaBoost are sophisticated boosting algorithms; each has been utilized across a multitude of tasks, encompassing classification and regression challenges within diverse sectors such as healthcare and finance. They have demonstrated robust performance on benchmark datasets and have frequently attained state-of-the-art outcomes in a variety of competitions and practical applications.

The results are displayed in Table 1, with the best result highlighted in bold. It can be observed from Table 1 that the RF model exhibits the best performance among these models, which is consistent with the findings in [11,32,33]. The performance for the model includes an error rate of 19.16%, an AUC_ROC value of 0.7390, an F1 score of 0.3380, a Micro-F1 score of 0.3462, and a Macro-F1 score of 0.3476 on average. Based on these results, we selected the RF model for PE risk prediction and follow-up analysis.

Elastic Net has also found numerous successful applications, including PE risk prediction [13]. The sample size in this study is significantly smaller, being less than one-tenth of that utilized in [13]; additionally, certain features exhibit high correlation, such as those related to blood pressure (diastolic and systolic pressure) and historical factors (pregnancy history, fertility history, PE history, family PE history), among others. The small sample size and serious feature correlation in the datasets in this study could diminish the advantages of the Elastic Net, increasing its vulnerability to overfitting and necessitating the complex optimization of multiple hyperparameters. These considerations were paramount in our decision not to compare our method with the Elastic Net approach. In addition, deep learning algorithms were not used for comparison due to the small sample size, which does not meet the extensive data demands typically associated with deep learning models.

### 2.2. PE Risk Prediction with Mono-Platform or Bi-Platform Data

The prediction model is trained on the Simoa Set and tested on the Elecsys Set, trained on the Elecsys Set and tested on the Simoa Set, and trained and tested on Bi-platform Fusion data for comparison. By setting the output threshold at 0.5 for decision making, we obtain results denoted by Simoa_Results, Elecsys_Results, and Simoa_Elecsys_Results, respectively, which are presented in the first three columns of Table 2 in the form of median ± deviation. Table 2 shows that the model trained from Bi-platform Fusion data outperforms those trained from each of the mono-platform data. It achieves the best performance with an error rate of 18.21%, an increased TPR of 25.35%, and a decrease in FPR of 1.85%. Additionally, there is an increase in AUC_ROC of 0.7610, AUC_PRC of 0.5348, and F1 score of 0.3850 compared to the results from Simoa and Elecsys.

The data imbalance between cases and controls in the Bi-platform Fusion data is significant. Setting the output threshold at 0.5 without considering this imbalance may not be appropriate, as the true positive rate (TPR) remains low, at only 25.35%. We adjust the threshold by performing 10-fold cross-validation on the dataset 100 times to achieve the best TPR performance, highlighting the final result in bold. The optimal output threshold is then searched using F1 as the performance evaluation index. The experimentally determined optimal threshold is 0.21 ± 0.0799. The performance at this threshold is presented in the fourth column of Table 2, revealing a significant increase in TPR to 71.47% and a slight increase in FPR to 15.19%. The AUC_ROC is 0.7627, AUC_PRC is 0.7271, and F1 score is 0.5520, indicating that the optimal threshold of 0.21 effectively addresses the case–control imbalance. The significantly better performance is shown in the prominently marked fourth column of Table 2.

### 2.3. Results on Test Set

The performance on Test_set, denoted by Simoa_Elecsys_Test_Results, is shown in the last column of Table 2 when the optimal output threshold of 0.21 is used.

When comparing the last two columns in Table 2, it is evident that the performance decreases in all the performance indices for the Test_set compared to the cross-validation performance on bi-platform data. While TPR and FPR remain acceptable, at 60.67% and 28.14%, respectively, they are not satisfactory. This indicates that the generalization of the trained model is still limited, partly due to the limitation of the training samples, as well as the potential distribution dissimilarity between the training data and test data. This situation provides an opportunity for further study.

### 2.4. Early PE Risk Prediction

Early pregnancy is the key period in which PE risk prediction is particularly important because it can introduce early intervention which will greatly reduce the incidence of PE and improve the health level of mother and fetus. The early pregnancy data in this study are First_Trimester Set, which come from the mono-platform SiMoA. While all available data for training a model are Simoa_Elecsys Set, we compare the model trained from only First_Trimester Set and that from Simoa_Elecsys Set to see their prediction performance on First_Trimester Set.

The model was trained using the First_Trimester Set, and the performance of 100 rounds of 10-fold cross-validation was recorded as First_Trimester_Results.

To train an early risk prediction model using the Simoa_Elecsys Set, 100 rounds of modified 10-fold cross-validation were conducted. Each round involved randomly dividing the First_Trimester Set into ten parts of approximately the same size. Nine parts were combined with the non-early pregnancy data in the Simoa_Elecsys Set for training, while the remaining part was used for testing. The performance results obtained at the output threshold of 0.21 were recorded as Simoa_Elecsys_Results2.

The results are presented in Table 3, with the best result highlighted in bold. It appears that the average performance of the model trained from the Simoa_Elecsys Set is superior to that of the model trained from the First_Trimester Set. The TPR improved from a less stable 57.14% to a more stable 69.9% (with a standard deviation of 15.47% decreasing to 13.80%), while the FPR decreased from 26.32% to a more stable 23.51%. Additionally, AUC_ROC increased from 0.7018 to a more stable 0.7627, and AUC_PRC increased from 0.5498 to a more stable 0.6544. The F1 score also increased from 0.4888 to a slightly less stable 0.5442 (a smaller standard deviation indicates greater stability of the corresponding median performance). These results suggest that utilizing all available data for modeling can help compensate for the lack of early pregnancy data and enhance early pregnancy PE risk prediction performance.

### 2.5. Feature Importance Ranking

Using the RF algorithm, feature importance was also obtained. The feature importance ranking results obtained by modeling Simoa Set, Elecsys Set, Simoa_Elecsys Set, and First_Trimester Set are shown in Figure 1. The top five features are colored red, the sixth to tenth are yellow, and the rest are green. The figure shows that the top five ranked features are PlGF, MAP, diastolic blood pressure, systolic blood pressure, and BMI, regardless of the data used for modeling. This ranking is also in line with the assessments made by local doctors. In the First_Trimester Set modeling, BMI before pregnancy is ranked within the top five, as opposed to current BMI, since there is no current BMI available during early pregnancy.

The importance of these features can be explained. During the development of PE, a low level of PlGF indicates insufficient placental angiogenesis or poor placental perfusion. The abnormal increase in systolic, diastolic blood pressure, and/or MAP reflects the disorder of systemic vascular function, which is closely related to the development and severity of PE. Additionally, high BMI causes metabolic and endocrine changes through more body fat accumulation, leading to increased insulin resistance and impaired vascular endothelial function, thus increasing the burden on blood vessels.

## 3. Discussion

Through this small sample and case–control imbalance problem, we compared the performance difference between the model trained from real data and the model trained from real + augmented data to see the effect of data augmentation on model performance.

### 3.1. Data Augmentation Using SMOTE-Based Algorithms

In view of imbalance on cases and controls, we used a SMOTE-based algorithm to augment the minority group to balance the number of samples in different groups. Borderline SMOTE [34], ADASYN [35], SMOTENC [36], SVM SMOTE [37], and K-Means SMOTE [38] algorithms are used to augment the data. Experimental results show that K-Means SMOTE, with data augmented at a ratio of 1.2 cases to controls, achieves the best performance. Therefore, the K-Means SMOTE algorithm with a ratio of 1.2 was chosen to augment the data.

To understand the effect of data augmentation on performance, we trained a model with the RF algorithm from real data (Elecsys Set) and that from real + augmented data. Experimental results are given in Table 4. In Table 4, the first and second rows show the standard cross-validation performance of the model trained from and tested on the real data, and on the real + augmented data, respectively; the third row is the modified cross-validation performance of the model trained from the real + augmented data and tested on only the real data (i.e., the real + augmented data are divided at random into 10 parts of equal size, with nine parts used for training, and only the real data in the remaining part used for testing).

### 3.2. Virtually High-Performance Phenomenon

Upon comparing the results of the first and second rows in Table 4, it is evident that the model’s performance significantly improves with the use of data augmentation. For instance, the average error rate decreases substantially from a less stable 12.95% to a more stable 6.06%. This indicates that data augmentation has a positive impact on the model’s performance.

However, this is an illusion, as can be seen from the comparison of the second and third rows of Table 4; the model trained from real + augmented data has worse median performance and the performance is more unstable when tested on real data, e.g., the average error rate is increased from 6.06% to the less stable 11.76%; F1 decreases from 0.9351 to the less stable 0.7577; and AUC_ROC decreases from 0.9602 to the less stable 0.8576. This brings a serious problem; performance on real + augmented data is virtually high and not stable.

Comparing the first and third rows of Table 4, which were both tested on real data, it is evident that data augmentation during model training improves little but leads to unstable performance, which is reflected in the Error Rate, F1 score, and AUC_ROC metrics.

What is demonstrated in Table 4 is that though the model is trained from real + augmented data, the performance on real + augmented data is an illusion and virtually high, which cannot reflect the performance on real data.

This phenomenon can also be seen in our experimental results on each of the four datasets, as shown in Table 5. In Table 5, the first/second number in each grid of the table represents the performance tested on real data for the model trained from real data/real + augmented data, respectively. Then, the difference between the first number and the second number in the grid reflects the performance change on real data by learning a model without and with data augmentation.

The tendency of increased/decreased median performance but decreased/increased performance deviation can be seen from Table 5. In other words, adding augmented data for training a model improves but destabilizes model performance. This indicates that data augmentation has limited ability in solving small sample and sample imbalance problems, which is the direction we are concerned about and will further investigate.

The virtually high-performance phenomenon was observed for not only K-Means SMOTE [38], but also borderline SMOTE [34], ADASYN [35], SMOTENC [36], and SVM SMOTE [37] algorithms.

In fact, SMOTE-based approaches cannot guarantee that the distribution of augmented data will be the same as that of the real data. This is due to its simple linear interpolation and ignorance of local density differences, leading to the possibility of amplifying noises and losing the diversity of real data. By using both real and augmented data to train a model, the model can effectively generalize to a combination of real and augmented data. However, it may struggle to generalize to real data alone due to potential overfitting on the augmented data.

To avoid the phenomenon, the distribution of augmented data should be as close as possible to that of real data. For this purpose, a GAN-based approach is believed to be a good solution to this problem.

## 4. Material and Methods

### 4.1. Quality Participant Selection

The data for this study were collected from hospitals in Xinjiang, China by the National Engineering Research Center for Miniaturized Detection.

Tens of thousands of samples were examined, and quality samples of PE and normal controls after maternity delivery were selected. Peripheral blood samples were sourced from retrospective studies, taken and tested at different stages of pregnancy, with relevant pregnancy information recorded.

Diagnostic criteria for PE were based on international guidelines provided in guidelines for the diagnosis and treatment of hypertension in pregnancy (2015) [15]. Women > 18 years of age with a singleton pregnancy were included if they presented with hypertension (systolic blood pressure [BP] > 140 mm Hg and diastolic BP > 90 mm Hg) and proteinuria (>1 + in dipstick, which corresponds to a protein concentration ≥ 30 mg/dL protein in spot urine) after 20 weeks of gestation. If there is no proteinuria but any of the following organs or systems are involved, the diagnosis is confirmed: heart, lung, liver, kidney, and other important organs; abnormal changes in the blood system, digestive system, and nervous system; and placenta–fetal involvement.

Participants in the control group were randomly selected from the hospitals during the same period. They had no history of chronic disease or pregnancy-related complications, and no abnormal blood pressure (blood pressure < 140/90 mmHg), premature rupture of membranes, placenta previa, or threatened miscarriage during pregnancy. They had no history of chronic hypertension, preeclampsia, diabetes, heart disease, liver disease, kidney disease, immunotherapy, or blood transfusion. They had full-term delivery with no history of fetal protection treatment or cervical cerclage during pregnancy. Their auxiliary examinations during pregnancy, including blood/urine routine, urine protein, liver and kidney function, were all normal.

By selecting quality participants, we obtained the datasets for this study, which include two independent sets: the Simoa Set and the Elecsys Set. PlGF in the former set was collected using the SiMoA human PlGF discovery kit and the SiMoA platform (Quanterix, Lexington, MA, USA), while PlGF in the latter set was collected using the ROCHE Elecsys cobas e 411 platform. The combination of the two sets is denoted as the Simoa_Elecsys Set.

The samples in the Simoa Set, which relates only to early pregnancy, is denoted as First_Trimester Set (there are too few early pregnancy samples in the Elecsys Set), where early pregnancy is 11 − 3 + 6 gestational weeks.

The test set is independently collected for evaluating the PE risk prediction model. It is from the SiMoA platform and has no intersection with any other sets.

The summary of these five datasets is given in Table 6. The datasets are of great case–control imbalance.

Each sample in the dataset includes 14 features, continuous (e.g., diastolic pressure, systolic pressure, PlGF) and discrete (e.g., nationality, family history of PE). The features of some samples were not recorded or were missing. On average, the missing data take up about 1/4 of the total size of a dataset for the mentioned datasets above.

All the datasets indicate that PE risk prediction is a typical small sample and sample imbalance problem, with missing data included, which is common in medical data analysis.

Furthermore, the PlGF Calibration Set, which includes a total of 24 samples, was collected. Only the PlGF values of these samples were measured from both the SiMoA and Elecsys platforms without any missing data. This set was gathered for studying the calibration of PlGF from either one of the two platforms.

### 4.2. Framework of Using Machine Learning Approaches

In this approach, we primarily use two types of machine learning models: MLP and RF. MLP is utilized for missing data imputation and bi-platform PlGF calibration, while RF is employed for PE risk prediction. RF outperforms typical models such as MLP, SVM, XGBoost, and AdaBoost in terms of performance.

The framework of our machine learning-based PE risk prediction is shown in Figure 2. It includes two phases: data preprocessing and risk prediction. The former includes feature encoding, missing data imputation, and PlGF calibration, and the latter is to construct risk prediction models from the preprocessed data.

### 4.3. Feature Encoding

For a sample, 14 features are collected: age, ethnicity, height, pre-pregnancy weight, current weight (current means at the time that the PE risk is tested, which relates to pregnancy week), pregnancy history, fertility history, PE history, family PE history, hypertension history, gestational week, diastolic pressure, systolic pressure, and PlGF.

Pre-pregnancy BMI, current BMI, and BMI increase rate are then derived from pre-pregnancy weight and current weight. Gestational day is used instead of gestational week; so is the mean arterial pressure (MAP) [39], calculated from diastolic pressure and systolic pressure; among the features, the non-digital feature “nationality” is divided into five categories:The Han ethnic group, Uygur ethnic group, Kazak ethnic group, Hui ethnic group and other ethnic groups, and is digitalized by the One-Hot Encoding technique [40]. Therefore, there are 22 features in total for a sample.

### 4.4. Missing Data Imputation

We propose a missing data imputation method that includes various strategies for adapting to training data and test data, as well as independent features and dependent feature pairs.

Training data contain class label information. With this information, for independent features, the intra-class median strategy is utilized (see Figure 3b); the intra-class median value of a feature is used to replace any missing values of that feature.

Here, median is used instead of mean due to its well-known property of being much more robust to noise and/or outliers compared to the mean. Intra-class imputation is proposed rather than conventional over-class imputation, since the feature is generally considered to be class-dependent; for example, blood pressure is generally high for cases and not high for controls. This approach makes the imputed value more reasonable compared to the conventional over-class imputation, as it utilizes class label information in the training data. For test data, no class information can be applied. A conventional strategy of over-class median is proposed for imputing the missing data of a feature for independent features for a test sample.

More precisely, suppose we have a training dataset $\Omega_{Train}=\{(x_{i},y_{i}):x_{i}\in R^{d},$$y_{i}\in \{0,1\}\;for\;i=1,2,\ldots ,N\}$ and a test dataset $\Omega_{Test}=\left\{{x_{i}}^{\prime }:{x_{i}}^{\prime }\in R^{d},for\;i=1,2,\ldots ,M\right\}$, where $x_{i}$ and ${x_{i}}^{\prime }$ are d-dimension feature vectors, with the jth feature being $x_{ij}$ and ${x^{\prime }}_{ij}$. The missing $x_{ij}$ in $\Omega_{Train}$ is imputed by
(1)$$\langle x_{ij}\rangle =\underset{k}{\mathrm{median}}\left\{x_{kj},if\;y_{k}=y_{i}\right\}$$
and the missing ${x^{\prime }}_{ij}$ in $\Omega_{Test}$ is imputed by
(2)$$\langle {x^{\prime }}_{ij}\rangle =\underset{k}{\mathrm{median}}\left\{x_{kj}\right\}$$

Among the features in this approach, two feature pairs are seriously dependent: current weight and pre-pregnancy weight, and diastolic pressure and systolic pressure. For both pairs, we propose a relation-based missing data imputation scheme by utilizing the relation between the features in each pair. The relation is given by MLPs trained with the non-missing values of the two features.

Specifically, if a feature value is missing in the case sample/control sample, it is imputed from the other dependent features, via their relation modeled by an MLP dependent on the case/control group. Only when both values are missing is the intra-class median adopted for training data and the over-class median adopted for test data.

More precisely, for a dependent feature pair, feature a and feature b, we use the feature pair-related training dataset ${Ʊ}_{Train}$ = $\left\{{(x}_{ia},x_{ib},y_{i}):x_{ij}\in R,j=\{a,b\},i=1,2,\ldots ,N\right\}$ and test dataset ${Ʊ}_{Test}=\left\{{(x^{\prime }}_{ia},{x^{\prime }}_{ib}):{x^{\prime }}_{ij}\in R,j=\{a,b\},i=1,2,\ldots ,M\right\}$, where $x_{ij}$ and ${x^{\prime }}_{ij}$, j = a, b are real numbers. The missing $x_{ia}$ and/or $x_{ib}$ in ${Ʊ}_{Train}$ is imputed by
(3)$$\langle x_{ia}\rangle =\left\{\begin{array}{c} \mathrm{MLP}1(x_{ib}),\;\mathrm{if}\;x_{ib}\;\mathrm{is}\;\mathrm{not}\;\mathrm{missing}\;\;\;\;\;\;\;\;\;\;\;\;\;\;\;\;\;\;\;\;\;\;\;\;\;\;\;\;\;\;\;\;\;\;\;\;\;\\ \underset{k}{\mathrm{median}}\left\{nonmissing\;x_{ka},if\;y_{k}=y_{i}\right\}\;\mathrm{if}\;x_{ib}\mathrm{is}\;\mathrm{missing} \end{array}\right.$$
(4)$$\langle x_{ib}\rangle =\left\{\begin{array}{c} \mathrm{MLP}2(x_{ia}),\;\;\mathrm{if}\;x_{ia}\;\mathrm{is}\;\mathrm{not}\;\mathrm{missing}\;\;\;\;\;\;\;\;\;\;\;\;\;\;\;\;\;\;\;\;\;\;\;\;\;\;\;\;\;\;\;\;\;\;\;\;\;\\ \underset{k}{\mathrm{median}}\left\{nonmissing\;x_{ka},\;if\;y_{k}=y_{i}\right\}\;\mathrm{if}\;x_{ia}\;\mathrm{is}\;\mathrm{missing} \end{array}\right.$$
(5)$$\langle x_{ij}\rangle =\underset{k}{\mathrm{median}}\left\{nonmissing\;x_{kj}\right\}\;\mathrm{if}\;\mathrm{both}\;x_{ib}\;\mathrm{and}\;x_{ia}\mathrm{are}\;\mathrm{missing}\;(\mathrm{for}\;j=a,b)$$
where the relationship model MLP1:$x_{\cdot b}\to x_{\cdot a}$ and the relationship model MLP2:$x_{\cdot a}\to x_{\cdot b}$ are both trained from the dataset ${\{Ʊ}_{Train}$: both $x_{\cdot b}$ and $x_{\cdot a}$ are not missing}. The missing data in ${Ʊ}_{Test}$ are then imputed by
(6)$$\langle {x^{\prime }}_{ia}\rangle =\left\{\begin{array}{c} \mathrm{MLP}1({x^{\prime }}_{ib}),\;\mathrm{if}\;{x^{\prime }}_{ib}\;\mathrm{is}\;\mathrm{not}\;\mathrm{missing}\;\;\;\;\;\;\;\;\;\;\;\;\;\;\;\;\;\;\\ \underset{k}{\mathrm{median}}\{{nonmissing\;x}_{ka}\},\mathrm{if}\;{x^{\prime }}_{ib}\mathrm{is}\;\mathrm{missing} \end{array}\right.$$
(7)$$\langle {x^{\prime }}_{ib}\rangle =\left\{\begin{array}{c} \mathrm{MLP}2({x^{\prime }}_{ia}),\;\mathrm{if}\;{x‘}_{ia}\;\mathrm{is}\;\mathrm{not}\;\mathrm{missing}\;\;\;\;\;\;\;\;\;\;\;\;\;\;\;\;\;\;\;\\ \underset{k}{\mathrm{median}}\{{nonmissing\;x}_{ka}\},\;\mathrm{if}\;{x’}_{ia}\;\mathrm{is}\;\mathrm{missing} \end{array}\right.$$
(8)$$\langle {x^{\prime }}_{ij}\rangle =\underset{k}{\mathrm{median}}\left\{nonmissing\;x_{kj}\right\}\;\mathrm{if}\;\mathrm{both}\;{x^{\prime }}_{ib}\;\mathrm{and}\;{x^{\prime }}_{ia}\;\mathrm{are}\;\mathrm{missing},\;\mathrm{for}\;j=a,b$$

We take pre-pregnancy weight and current weight as an example. Current weight is related to and varies with the feature of gestational week. For this reason, the gestational week feature is divided into five intervals, namely 11–13 weeks, 14–18 weeks, 19–23 weeks, 24–28 weeks, and 29–33+6 weeks, to reduce the effect of the gestational week on missing data imputation. For each gestational week interval, two MLPs are trained. Specifically, denote non-missing data in the feature of pre-pregnancy weight and current weight as D1 and D2, respectively. They are used for training two models MLP1 and MLP2 to represent the relation from D1 to D2 and that from D2 to D1, respectively. Using D1 as input and D2 as output, an MLP model denoted as MLP1 is trained to impute the missing current weight, if it is missing, from the feature of pre-pregnancy weight. Conversely, utilizing D2 as input and D1 as output, another MLP model denoted as MLP2 is trained to impute the pre-pregnancy weight, if it is missing, from the feature of current weight.

The use of either MLP1 or MLP2 depends on the actual missing situation (see Figure 3a). The intra-class median of each feature is adopted only when both pre-pregnancy weight and current weight are missing.

Min-max normalization is conducted on each feature, except for PlGF.

### 4.5. PlGF Calibration

PlGF Calibration is a regression problem. Two MLPs are trained, and the better one is used for calibration. For the PlGF Calibration Set, denote the PlGF values of the two platforms by D_Simoa_ and D_Elecsys_, respectively. Two MLPs are trained from the set; D_Simoa_ is used as input to fit D_Elecsys_ with an MLP denoted by MLP_3_, and D_Elecsys_ as input to fit D_Simoa_ with an MLP denoted by MLP_4_, as shown in Figure 4a.

Because there are only 24 samples available for PlGF calibration, both MLPs are designed to be relatively simple. Each MLP consists of one input, two hidden neurons, and one output neuron. The activation functions used are sigmoidal for the hidden neurons and linear for the output neuron. To ensure reliable performance, 100 rounds of 3-fold cross-validation are conducted.

Calibration is significant for subsequent PE risk prediction, so the calibration model must be sufficiently reliable. To ensure this, the model with the median performance out of the 300 established models is chosen.

Specifically, we represent the calibration data by $\left\{(s_{i},e_{i}):i=1,2,\ldots ,24\right\}$, within which there are only 24 samples whose PlGF is measured by SiMoA and by the Elecsys platform, being $s_{i}$ and $e_{i}$ for the ith sample. We establish the regression model MLP3:s→e and MLP4:e→s in the same method.

We use MLP3 as an example to demonstrate our method. Through 100 rounds of 3-fold cross-validation, we establish and evaluate 300 models (using MSE). Then, the model MLP3 is set to be the $M_{median}$ whose performance is
(9)$$P_{median}=median\{P_{i},i=1,2,\ldots ,300\}$$

The idea behind this concept comes from the Central Limit Theory, which states that in certain conditions, the sum of a large number of independent random variables approaches a Gaussian distribution. Therefore, the mean of the distribution is considered the most reliable performance compared with all the other ones. We use the median instead of the mean to account for performance outliers, making the median performance, or more specifically, the model with the median performance, a more reliable indicator compared to others.

By comparing the obtained mean square error of MLP3 and MLP4, denoted by MSE3 and MSE4, respectively, it is found that MSE4 < MSE3. Therefore, the PlGF value measured in the Elecsys platform is predicted by MLP4 for calibration to the SiMoA platform, while there is no need for calibration for the value obtained from the SiMoA platform, shown in Figure 4b.

### 4.6. PE Risk Prediction

Models trained by a spectrum of typical machine learning algorithms from various datasets are experimentally compared to establish the best-performing PE risk prediction model and early prediction model. The various datasets used for training are the data with PlGF from the mono-platform vs. those from the bi-platform, the early pregnancy data vs. early plus non-early pregnancy data, and the dataset from real data vs. from real plus augmented data. The best model is then applied to the independent test set.

The prediction model can accept samples whose PlGF is collected from either SiMoA or the Elecsys platform, and the model then outputs the PE risk of the sample.

## 5. Conclusions

We propose a PE risk prediction approach based on machine learning algorithms. The approach is substantially different from existing ones. However, conducting a comparative study with existing methods is challenging due to the lack of a model allowing PlGF measured from either SiMoA or the Elecsys platform, as well as a much smaller data size compared to other approaches (less than one-tenth of that in [13]). The distinct features of this study are as follows. (1) It provides a novel imputation method and a novel calibration method for imputing missing data and reliably calibrating the PlGF from the two platforms; (2) a large number of rounds of cross-validations are conducted for evaluating a model on not only its performance but also the stability of the performance; (3) median model and/or median performance is adopted for robustness to outlier model and/or outlier performance; (4) models trained from various datasets, mono-platform vs. bi-platform data, early pregnancy vs. early plus non-early pregnancy data, real vs. real plus augmented data, are compared for the best model, to address the small sample problem; (5) experiments on the effect of data augmentation demonstrate the detrimental potential of achieving virtually high but unstable performance. This phenomenon is undesirable and warrants the attention of the machine learning community.

From experiments, we have drawn the following conclusions. (1) The model trained using Fusion data collected from both SiMoA and Elecsys platforms achieves higher PE risk prediction performance compared to the model trained using data from only one platform; (2) non-early pregnancy data can help address the limitation of early pregnancy data and improve early pregnancy PE risk prediction performance; (3) optimizing the output threshold for decision making effectively addresses case–control imbalance; (4) data augmentation with SMOTE-based algorithms does not improve prediction performance on real data, despite showing high performance on real + augmented data.

This study is effective. From the prospective observations based on the model of this study, the incidence of PE in hospitalized pregnant women in Xinjiang, China, was reduced from 7.2% to 2.0%, with the mortality rate reduced to 0.

The method proposed in this paper is intended for application in PE risk prediction in a broader region of China. Data will be collected in the region to train the models in this study more effectively, speeding up the training process and achieving a generalizable model with predictive power. This method can also be utilized for other disease risk prediction and diagnosis applications, as well as for applications where machine learning methods are employed.

The limitations of this approach and challenges of machine learning are:(1)Missing data: This approach, akin to all other machine learning methodologies, employs certain criteria to impute missing data. However, the imputed data may not be, and generally are not, the true values, which has an impact on subsequent studies. Exploring machine learning methods suitable for datasets with missing data directly, rather than relying on the imputation of missing data, is of great significance to ensure that subsequent studies are not skewed.(2)Small samples and sample imbalance: The number of PlGF calibration samples is only 24, and the other datasets in this study are also small in size. The consequence of small samples may lead to an unreliable model for PlGF calibration, PE risk prediction, and early prediction. In addition, the datasets are seriously imbalanced between cases and controls. In this approach, we adopted a simple output thresholding approach; however, determining the threshold is not trivial.(3)Intrinsic feature discovery: The RF model has been found to perform the best among other models in PE risk prediction. It utilizes all the currently available features in this research, but some of these features may be highly correlated and more important for the prediction. For instance, among the top five features ranked by importance, MAP, diastolic blood pressure, and systolic blood pressure are related to blood pressure and are believed to be interrelated, implying that the RF approach cannot provide the intrinsic features of the problem nor utilize them for risk prediction. This indicates the existence of a better model than the RF model. Only the model trained using the intrinsic features of the problem can achieve the best performance.(4)Reliable model construction: This research utilizes multiple rounds of 10-fold cross-validation. This method is solely used for evaluation purposes. Compared to a single instance of 10-fold cross-validation, it can assess model performance more accurately and reliably. However, it is not a means to enhance model performance. How to train the model from a limited dataset to achieve highly reliable, stable, and high performance remains a challenge to the machine learning community.(5)Omics data utilization: This approach utilizes only a limited biomarker of PE, the PlGF. However, PE is a complex pregnancy disorder with phenotypes characterized by clinical signs and symptoms that may have genetic underpinnings. It requires the use of omics data along with the symptoms for risk prediction, especially early risk prediction. PE has a genetic predisposition, yet it is not a traditional single-gene inherited disease, indicating that genetic factors, along with environmental and lifestyle factors, play a role in pathogenesis.

Due to the above limitations and challenges, future work may include, but is not limited to, the following directions: collecting more quality data to refine the models in this study, utilizing state-of-the-art GAN-based techniques for data augmentation to address small sample sizes and sample imbalances, expanding the study to include omics data, refining feature analysis with advanced computational methods to enhance predictive power, and exploring a deep learning framework driven by small samples to improve model predictability, stability, and reliability.

## Figures and Tables

**Figure 1 ijms-25-10684-f001:**
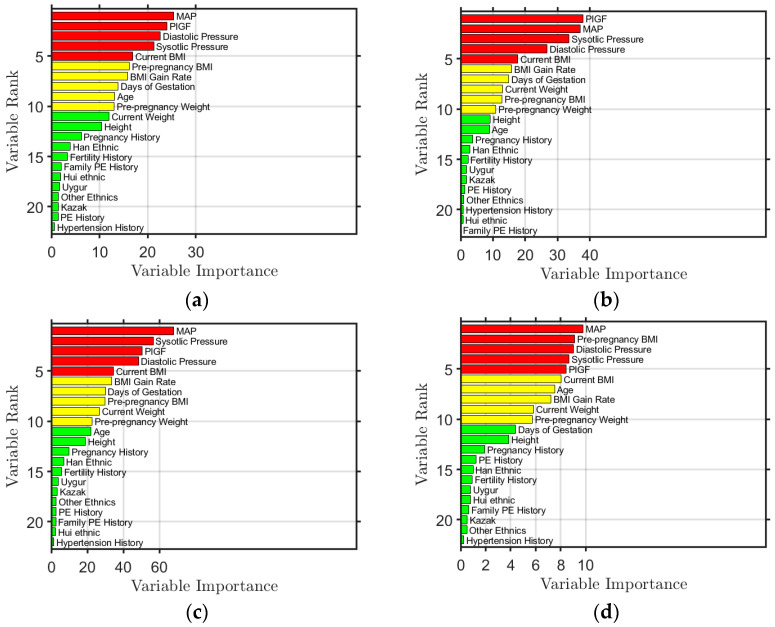
Ranking of feature importance obtained from the model trained from (**a**) Simoa Set, (**b**) Elecsys Set, (**c**) Simoa_Elecsys Set, and (**d**) First_Trimester Set, where the features ranked in the top 5 are colored red, the features ranked from 6th to 10th are colored yellow, and those ranked from 11th to 22nd are colored green.

**Figure 2 ijms-25-10684-f002:**
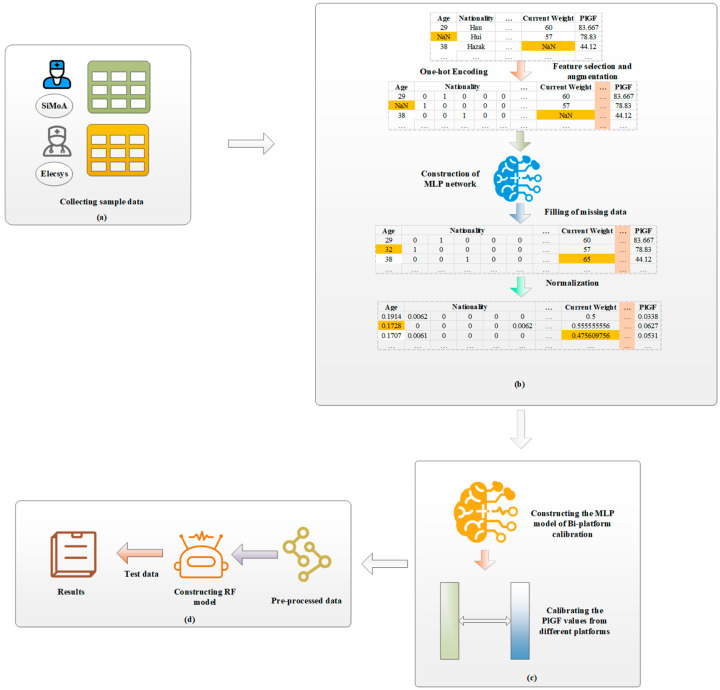
Framework of PE risk prediction based on RF and bi-platform calibration. (**a**) Collecting PE case group sample data and control group sample data; (**b**) coding features, imputing missing data with MLP networks, and normalizing features; (**c**) calibrating PlGF from the two platforms with an MLP model; and (**d**) constructing PE risk prediction model and predicting PE risk of test samples.

**Figure 3 ijms-25-10684-f003:**
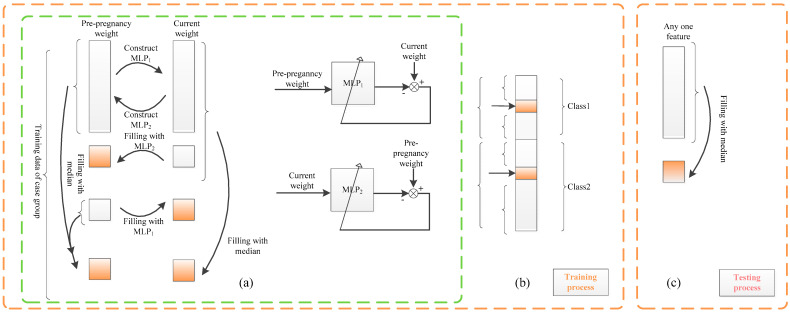
Missing data imputation based on MLP (missing data are represented by imputed orange). (**a**) The training process: Take the pair of pre-pregnancy weight and current weight of case group as an example. (**b**) Training process: Other features are imputed by intra-class median. (**c**) Test process: The missing data of other features are imputed by the median.

**Figure 4 ijms-25-10684-f004:**
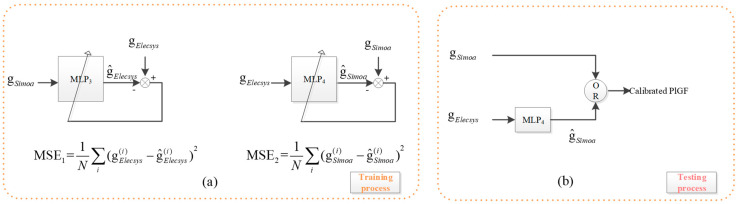
PlGF value calibration based on MLP. Since MSE_2_ < MSE_1_, the PlGF values detected from the SiMoA platform do not need to be calibrated, while the PlGF values detected from the Elecsys platform are to be calibrated with MLP_4_.

**Table 1 ijms-25-10684-t001:** Performance evaluation using different machine learning algorithms on Simoa Set.

Algorithm	Error Rate	F1	Micro-F1	Macro-F1	AUC_ROC
MLP	0.2270 ± 0.0260	0.2302 ± 0.1225	0.2685	0.2644	0.6631 ± 0.1020
SVM	0.2270 ± 0.1203	0.2353 ± 0.1040	0.2677	0.3129	0.6994 ± 0.1169
RF	0.1916 ± 0.0373	0.3380 ± 0.1293	0.3462	0.3476	0.7390 ± 0.0561
XGBoost	0.2127 ± 0.0390	0.3205 ± 0.1137	0.3605	0.3676	0.7387 ± 0.0495
AdaBoost	0.1986 ± 0.0288	0.3333 ± 0.1275	0.2197	0.2104	0.7245 ± 0.0780

**Table 2 ijms-25-10684-t002:** Performance of the RF model trained from and tested on a mono-platform dataset, Bi-platform Fusion Set (output threshold being set at 0.5), Bi-platform Fusion Set (threshold being the optimal 0.21), and trained from Bi-platform Fusion Set and tested on Test_set.

Performance	Simoa_Results	Elecsys_Results	Simoa_Elecsys_ Results (Threshold = 0.5)	Simoa_Elecsys_ Results (Threshold = 0.21)	Simoa_Elecsys_ Test_Results (Threshold = 0.21)
Error Rate	0.2133	0.2031	0.1821 ± 0.0618	0.1964 ± 0.0976	0.2955
True positive rate (TPR)	0.1243	0.1034	0.2535 ± 0.0153	0.7147 ± 0.1344	0.6067
False positive rate (FPR)	0.0000	0.0233	0.0185 ± 0.0153	0.1519 ± 0.1302	0.2814
AUC_ROC	0.7051	0.6902	0.7610 ± 0.1056	0.7627 ± 0.1019	0.7092
AUC_PRC	0.6556	0.1652	0.5348 ± 0.1094	0.7271 ± 0.1840	0.6851
F1	0.2211	0.1734	0.3850 ± 0.2441	0.5520 ± 0.1302	0.3435

**Table 3 ijms-25-10684-t003:** Comparison of bi-platform fusion model and early pregnancy model (at output threshold of 0.21).

Performance	First_Trimester_Results	Simoa_Elecsys_Results2
Error Rate	0.3915 ± 0.1003	0.2536 ± 0.0680
True positive rate (TPR)	0.5714 ± 0.1547	0.6990 ± 0.1380
False positive rate (FPR)	0.2632 ± 0.0831	0.2351 ± 0.0626
AUC_ROC	0.7018 ± 0.1222	0.7627 ± 0.1019
AUC_PRC	0.5498 ± 0.2010	0.6544 ± 0.1522
F1	0.4888 ± 0.0908	0.5442 ± 0.1134

**Table 4 ijms-25-10684-t004:** Effect of data augmentation on model performance (results on Elecsys Set).

Prediction Performance	Error Rate (%)	F1	Micro-F1	Macro-F1	AUC_ROC
Model trained from and tested on real data	12.95 ± 4.06	0.6653 ± 0.1144	0.7004	0.6871	0.8261 ± 0.0717
Model trained from and tested on real + augmented data	6.06 ± 2.75	0.9351 ± 0.0284	0.9244	0.9252	0.9602 ± 0.0220
Model trained from real + augmented data and tested only on real data	11.76 ± 5.27	0.7577 ± 0.1027	0.7027	0.7131	0.8576 ± 0.0563

**Table 5 ijms-25-10684-t005:** Effect of data augmentation on model performance (results on the four datasets).

Prediction Performance Change	Simoa_Results	Elecsys_Results	Simoa_Elecsys_Results	First_Trimester_Results
Error Rate	0.1914 ± 0.0188 0.1838 ± 0.0449 ↑↓	0.1295 ± 0.0406 0.1176 ± 0.0527 ↑↓	0.1619 ± 0.0223 0.1490 ± 0.0304 ↑↓	0.2115 ± 0.0805 0.2111 ± 0.0816 ↑↓
True Positive Rate (TPR)	0.2762 ± 0.0773 0.2762 ± 0.1507 = ↓	0.5588 ± 0.1316 0.6396 ± 0.1229 ↑↑	0.4123 ± 0.0861 0.4616 ± 0.0935 ↑↓	0.3810 ± 0.2118 0.3875 ± 0.1920 ↑↑
False Positive Rate (FPR)	0.0536 ± 0.0203 0.0536 ± 0.0272 = ↓	0.0189 ± 0.0279 0.0381 ± 0.0374 ↓↓	0.037 ± 0.0214 0.0413 ± 0.0166 ↓↑	0.0526 ± 0.0666 0.0789 ± 0.0704 ↓↓
AUC_ROC	0.7732 ± 0.0445 0.8056 ± 0.0642 ↑↓	0.8261 ± 0.0717 0.8576 ± 0.0563 ↑↑	0.8170 ± 0.0412 0.8368 ± 0.0457 ↑↓	0.7500 ± 0.1172 0.7727 ± 0.1451 ↑↓
AUC_PRC	0.3885 ± 0.1312 0.3765 ± 0.2562 ↓↓	0.7807 ± 0.1014 0.7935 ± 0.1114 ↑↓	0.6173 ± 0.1159 0.7256 ± 0.1103 ↑↑	0.5971 ± 0.3041 0.5756 ± 0.3054 ↓↓
F1	0.3636 ± 0.0974 0.3693 ± 0.1598 ↑↓	0.66533 ± 0.1144 0.75770 ± 0.1027↑↑	0.5192 ± 0.0810 0.5891 ± 0.1039 ↑↓	0.4000 ± 0.1443 0.5227 ± 0.1729 ↑↓

In addition to the second number, two symbols are shown: ↑, ↓, or =. The first symbol indicates whether better (↑), worse (↓), or equal (=) median performance is achieved, while the second symbol indicates whether the performance is more (↑) or less (↓) stable when data augmentation is applied. The symbol represents the change in performance rather than the change in value.

**Table 6 ijms-25-10684-t006:** A brief description of five datasets.

	Simoa Set	Elecsys Set	Simoa_Elecsys Set	First_Trimester Set	Test_Set
# cases	145	169	314	65	130
# controls	559	525	1084	190	892
# samples	704	694	1398	255	1022

## Data Availability

The raw data supporting the conclusions of this article will be made available by the authors on request.

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
