# Peer review of "A Prospective Study on Risk Prediction of Preeclampsia Using Bi-Platform Calibration and Machine Learning"

_ijms, 2024, doi:10.3390/ijms251910684_

Round 1
Reviewer 1 Report
Comments and Suggestions for Authors
The paper is not technicaly properly processed:
-The abstract has 229 words while the maximum is 200 words also there are abbrevations what is not acceptable. The abstract should must summarize article's with the clear explantation of the subject, aim and scientiffic contribution of the paper.
-The paper is not well organized and because of that difficult to understand.
The section 1. Introduction should clear highlight what is it's the the subject, aim and paper significance. Literature review is not presented in this section or separate subsection or section. It shuld has and short overview of planed section organization in which the authors will present theirs research.
Chapter named 2.Material and Methods should be before Chapter 3.Results instead of now in the paper of the named section 2. Research design and Methods and with suitable subsections
All sections inthe paper shold be numbered as well as it was done with the subsections in the paper. The tables must be properly marked and Figure 4. Should be clearer.about used material and methods
- The subject matter is not presented in a comprehensive manner:
The authors did not explain why they chose those “typical“ machine learning algorithms, multilayer perceptron (MLP), Support Vector Machine (SVM), Random Forest (RF), XGBoost, and AdaBoost, for the best performing model for PE risk prediction.
The authors had not explain how imbalnced nature of considered dataset impact on choosing adequate measure for measuring the quality of proposed machine learning algorithms and using PRC measure.
The paper lacks with limitations of the proposed framework and comparation proposed framework with already existing.
- Also, the number from total 34 references in the paper is too small for the publication of the paper in such an eminent journal as it it is IJMS.
Author Response
请参阅附件。

Reviewer 2 Report
Comments and Suggestions for Authors
1.In the introduction, the background, purpose, and significance of the research can be more clearly explained to highlight innovation and advantages. It can emphasize the challenges faced by current PE risk prediction and how the proposed research methods can overcome them and improve prediction performance.
2.A review of the literature on existing PE risk prediction methods can be included, especially about research progress in recent years.
3.In the data collection and processing section, more details can be added, such as the data collection period, sample screening criteria, and specific data pre-processing steps.
4.More analysis and discussion can be added to the experimental results section, such as comparing the importance of different features in prediction and their biological explanations; analyzing the impact of data augmentation on model performance, and exploring its possible mechanisms.
5.It is necessary to improve the innovation of the method, and mathematical formulas can be added.
6.The latest comparison methods can be included.
7.Discuss the limitations of the model and future directions for improvement.
8.The conclusion section can summarize the research findings more concisely and clearly and indicate the innovative points and practical application value of the research.
9.TABLE IV does not display everything.
Comments on the Quality of English Language
Language expression requires proofreading to ensure correct grammar and clear expression.
Author Response
请参阅附件。

Reviewer 3 Report
Comments and Suggestions for Authors
The paper "A Prospective Study on Risk Prediction of Preeclampsia Using Bi-platform Calibration and Machine Learning" looks at using two different data systems and several machine learning methods to better predict preeclampsia. The study uses a large amount of data and new methods to try to improve predictions and health outcomes. However, the paper could be clearer in several areas. It needs more detailed explanations of how data was collected and how models were built. It should also explain why specific machine learning options were chosen and give a better description of the data used. Additionally, the paper should use simpler and clearer language to make it easier to read and understand. This would make the study's results clearer and more useful.
- Introduction: This part has been subdivide into too many paragraphs. Merge some of them if necessary.
"The cause of PE is still unknown and may involve a variety of maternal, placental and fetal factors [1]. It is believed to involve abnormalities in the placenta, leading to systemic inflammation, endothelial dysfunction, and impaired blood vessel function."
- While mentioning PE's multi-factor nature, it doesn't cover current main theories about its cause. Briefly introduce main academic views on PE causes, like sFlt-1/PlGF imbalance, to deepen the discussion.
“In this section, multiple machine learning algorithms are used to learn PE prediction models, and the best performing RF model is adopted …”
- This sounds like Methods. Suggest providing detailed performance comparison data instead.
"For constructing PE risk prediction system, more than tens of thousands of samples were tested..."
- No specific sample size given. Sample selection criteria not fully explained.
"PlGF levels, collected from SiMoA or Elecsys platform, are calibrated using simple MLP..."
- MLP used to calibrate PlGF levels from different platforms, but calibration model setup and validation not detailed. This step is key for data quality due to possible platform differences.
"However, this is an illusion, as can be seen from the comparison of the second and the third row of Table V(a)..."
- The author says performance boost from data augmentation is false, but doesn't explain why. Please consider discussing reasons for this false improvement, like data leakage or overfitting, and how to avoid it.
"This study is effective. From the prospective observations based on the model of this study, the incidence of PE in hospitalized pregnant women in the district of Xijiang, China, was reduced from 7.2% to 2.0%..."
- Discuss possible limits of these findings, like study design bias or regional healthcare differences. Also, mention plans to test the model in other areas or populations.
"the average error rate is greatly decreased from not so stable 12.95% to the more stable 6.06%."
- The terms "not stable" and "more stable" need quantitative support.
"From the prospective observations based on the model of this study, the incidence of PE in …”
- The sudden introduction of a specific case study feels abrupt.
Other comments:
- Use same abbreviations or full terms throughout. If "RF" (Random Forest) is defined first time, use the short form after.
- Keep decimal places consistent for performance metrics in Results. If error rate is given to two decimal places (like 19.16%), stick to this.
“Seen from Table III is that the model …”
- The model trained with Bi-platform Fusion data outperforms the models trained with mono-platform data, as shown in Table III.
Comments on the Quality of English Language
The language need major improvement. Simplifying and streamlining the language and ensuring consistent terminology would also enhance the manuscript’s readability and impact.
Round 2
Reviewer 1 Report
Comments and Suggestions for Authors
The authors accepted all my suggestions.
Reviewer 2 Report
Comments and Suggestions for Authors
Accept for publication.